# Global Voices, Local Biases: Socio-Cultural Prejudices across Languages

**Anjishnu Mukherjee**[*]   **Chahat Raj**[*]   **Ziwei Zhu**   **Antonios Anastasopoulos**
Department of Computer Science, George Mason University
{amukher6,craj,zzhu20,antonis}@gmu.edu

## Abstract

Human biases are ubiquitous but not uniform: disparities exist across linguistic, cultural, and societal borders. As large amounts of recent literature suggest, language models (LMs) trained on human data can reflect and often amplify the effects of these social biases. However, the vast majority of existing studies on bias are heavily skewed towards Western and European languages. In this work, we scale the Word Embedding Association Test (WEAT) to 24 languages, enabling broader studies and yielding interesting findings about LM bias. We additionally enhance this data with culturally relevant information for each language, capturing local contexts on a global scale. Further, to encompass more widely prevalent societal biases, we examine new bias dimensions across toxicity, ableism, and more. Moreover, we delve deeper into the Indian linguistic landscape, conducting a comprehensive regional bias analysis across six prevalent Indian languages. Finally, we highlight the significance of these social biases and the new dimensions through an extensive comparison of embedding methods, reinforcing the need to address them in pursuit of more equitable language models.[1]

## 1 Introduction

Language models, trained on large text corpora, have been shown to reflect and often exacerbate the societal perceptions found therein (Weidinger et al., 2021). As a result, there is a potential of harm (Bender et al., 2021; Kumar et al., 2023), e.g., due to the reinforcement of unfair representations of certain groups, especially in downstream usage of the representations from these models.

Despite the already growing interest in bias identification and mitigation in language representations, we identify several major shortcomings in the literature. First, the vast majority of the efforts on bias identification and mitigation in language representations, starting with the Word Embedding Association Test (WEAT; Caliskan et al., 2017), have been solely conducted on English. Some recent studies have also delved into other languages like Arabic, French, Spanish, and German, primarily by adapting WEAT to the languages (Lauscher and Glavaš, 2019) and integrating limited cultural contexts for non-social human biases (España-Bonet and Barrón-Cedeño, 2022), but still remain centered on the global North. Second, even though NLP has largely moved towards contextualized representations from transformer models like BERT, most works for determining bias in word embeddings have predominantly studied static encoding methods like FastText (Joulin et al., 2016) and GLoVe (Caliskan et al., 2022). To address these challenges, our work makes 5 contributions:

First, we construct a comprehensive dataset and perform in-depth analyses comparing machine vs. human translation for 24 non-English languages, encompassing previously overlooked ones from India and the global South. Additionally, we incorporate *language-specific* culturally relevant data wherever possible for both human and socio-cultural biases, covering all relevant categories in WEAT, a test for identifying biases based on the relative proximity of word embeddings in latent space by comparing associations with different classes of social biases. The languages in our dataset include Arabic (ar), Bengali (bn), Sorani Kurdish (ckb), Danish (da), German (de), Greek (el), Spanish (es), Persian (fa), French (fr), Hindi (hi), Italian (it), Japanese (ja), Korean (ko), Kurmanji Kurdish (ku), Marathi (mr), Punjabi (pa), Russian (ru), Telugu (te), Thai (th), Tagalog (tl), Turkish (tr), Urdu (ur), Vietnamese (vi), and Mandarin Chinese (zh).

Moving beyond English requires solutions to often ignored language-specific challenges. Our study includes the analysis of multi-word expres-

---

[*]Equal contribution
[1]All code, data and results are available here: https://github.com/iamshnoo/weathub.

sions corresponding to a single English word, ubiquitous in languages like Vietnamese or Thai. We also reformulate the WEAT metric to account for translation ambiguities in a principled manner.

Recognizing the dynamic nature of societal biases, we next observe that certain prevalent prejudices remain unaddressed in the existing WEAT categories. To remedy this, we propose five dimensions of human-centered biases that reflect contemporary ideologies and stereotypes (Mei et al., 2023), including toxicity, ableism, sexuality, education, and immigration.

In addition, we go beyond examining static embeddings and conduct a comprehensive analysis comparing various techniques for extracting word embeddings from transformer models. We also investigate the impact of employing a multilingual model (Levy et al., 2023) trained on a corpus of 100+ languages vs. monolingual models to understand which approach is better suited for capturing societal biases within different cultural contexts.

The final contribution of this work is an in-depth analysis of bias in seven Indian languages. We contrast multilingual and monolingual models and delve into relevant dimensions of social bias in India, inspired by Malik et al. (2022). Our experiments investigate gender bias in grammatically gendered languages such as Hindi vs. neutral ones like Bengali, alongside examining biases against lower castes, Islam, and professions associated with India's rural populace.

## 2 Data

To measure bias in a multilingual setting, we introduce WEATHub, a dataset that extends relevant WEAT categories to 24 languages, including cultural context, where applicable. Further, we include five new *human-centered* dimensions to capture the multifaceted nature of contemporary biases.

### 2.1 WEATHub: Multilingual WEAT

Building on the 10 tests for measuring word embedding associations between different groups of English words by Caliskan et al. (2017), we create a dataset of target and attribute pairs for evaluating different dimensions of bias in 24 languages.

We exclude tests involving target or attribute words specific to European American or African American names, as they lack relevance for non-Western and non-European languages. Furthermore, directly creating generalized translations of

**WEAT IDs and Target Attribute Pairs**

1. Flowers/Insects (Pleasant/Unpleasant)
2. Instruments/Weapons (Pleasant/Unpleasant)
6. Male/Female Names (Career/Family)
7. Math/Art (Male/Female Terms)
8. Science/Art (Male/Female Terms)
9. Mental/Physical Disease (Temporary/Permanent)

Table 1: WEATHub for assessing biases across 24 languages using six target-attribute pairs.

these names for different languages is challenging, given that European American versus African American dichotomy is not particularly pertinent in other languages. We also omit WEAT 10, which requires names of older and younger individuals, as it is a vaguely defined category that may not generalize well across languages. Consequently, our multilingual dataset for 24 languages focuses on WEAT categories 1, 2, 6, 7, 8, and 9 (Table 1), an approach similar to that employed by Lauscher and Glavaš (2019).

To collect annotated data in various languages, we provide our annotators with the English words and their corresponding automatic translation, separated by WEAT category. We provide instructions to verify the accuracy of the translations and provide corrected versions for any inaccuracies. Additionally, we ask annotators to provide grammatically gendered forms of words, if applicable, or multiple translations of a word, if necessary. All annotators who participated in our study are native speakers of their respective languages and have at least college-level education background. We also request that annotators provide five additional *language-specific* words per category to incorporate even more language context in our measurements of multilingual bias. For example, our Greek annotator added "anemone," while our Danish annotator added "chamomile" and "rose hip" for the WEAT category of flowers.

### 2.2 Human-centered Contemporary Biases

Bias is as pervasive and varied as the human experience itself. While some biases, such as the positive association with flowers and the negative connotations linked with insects, may be universal, other forms are far more complex and nuanced. Notably, the biases encountered by the LGBTQ+ community, immigrants, or those with disabilities are neither simple nor universally recognized.

Drawing upon research on intersectional biases (Tan and Celis, 2019; Hassan et al., 2021; Magee

| Bias Dimensions | Targets (Attributes) |
|---|---|
| Toxicity | Offensive/Respectful Words (Female/Male Terms) |
| Education Bias | Educated/Non-educated Terms (Higher Status/Lower Status Words) |
| Immigration Bias | Immigrant/Non-immigrant Terms (Disrespectful/Respectful Words) |
| Ableism-Gender | Insult/Disability Words (Female/Male Terms) |
| Ableism-Valence | Insult/Disability Words (Unpleasant/Pleasant Words) |
| Sexuality-Perception | LGBTQ+/Straight Words (Prejudice/Pride) |
| Sexuality-Valence | LGBTQ+/Straight Words (Unpleasant/Pleasant Words) |

Table 2: Five target-attribute pairs in WEATHub for human-centered contemporary bias assessment in 24 languages.

et al., 2021; Kirk et al., 2021; Parrish et al., 2022; Elsafoury, 2022; Mei et al., 2023), we propose five new dimensions - *Toxicity*, *Ableism*, *Sexuality*, *Education*, and *Immigration* bias. These are designed to fill in the gaps by explicitly targeting the assessment of social biases not currently captured within the existing WEAT dimensions. Table 2 delineates all the targets and attributes that gauge associations between different social groups.

The toxicity bias dimension tests the hypothesis that words associated with femininity are more likely to be linked with offensive language than those related to masculinity. Similarly, ableism bias seeks to discern prevalent biases against individuals with disabilities, examining if they are more frequently subject to derogatory language when their disability is mentioned. Meanwhile, the sexuality bias dimension examines the potential biases associated with different sexual orientations, specifically the use of LGBTQ+ and straight (cisgender, heterosexual and other non-LGBTQ+) words. The education bias analyzes educated and non-educated terms against a backdrop of higher and lower-status words to detect societal stereotypes about one's education. Last, the immigration bias dimension assesses whether immigrants are treated equally across different cultural contexts. While these dimensions were decided based on previous research and experiences of selected native speakers of different languages, we agree that framing biases under one general umbrella is difficult. We thus additionally frame the Ableism and Sexuality dimensions in terms of word associations with pleasantness (Omrani Sabbaghi et al., 2023) to offer a more intuitive angle for the problem.

To summarize our methodology for selecting the proposed dimensions: (1) We start with insights from intersectional biases research. (2) Then we engage in discussions with native speakers of the

24 languages establishing a foundational consensus on bias dimensions across linguistic communities. (3) Subsequently, we formulate the corresponding target/attribute pairs after weighing different options. (4) Finally, we opt for a naming convention that we felt best depicts these biases.

To establish the vocabulary used to measure associations, we created a basic set of words in English, building upon an initial lexicon proposed by the authors by querying web-based resources and various open-source and closed-source large language models. We then conversed with our annotators regarding the relevance of these lexical items across diverse cultural contexts, eventually selecting a set of ten words per category that demonstrate the highest degree of cross-cultural applicability. For the valence measurements, we re-used the pleasant and unpleasant words available from original WEAT categories 1 and 2 (Table 1). All of the word lists are available in our code repository.

## 2.3 India-Specific Bias Dimensions

Malik et al. (2022) provide an interesting case study for assessing bias in Hindi language representations, explicitly relating to various dimensions of bias pertinent to the Indian context. For instance, the caste or religion of an individual can often be inferred from their surnames, suggesting that associations may exist between a person's surname and specific target attributes such as descriptive adjectives or typical occupations. Furthermore, the grammatical gender inherent in Hindi necessitates investigating the associations between gendered words and male or female terms. We broaden these dimensions of bias to encompass seven diverse Indian languages and sub-cultures: English, Hindi, Bengali, Telugu, Marathi, Punjabi, and Urdu. Our study facilitates examining whether biases observed in Hindi persist in other Indian languages.

## 3 Methods

To overcome challenges arising from multilinguality, including but not limited to multi-word expressions and the absence of one-to-one translations from English to many languages, we reformulate the test statistic for WEAT. We also give a heuristic to help choose among multiple embedding methods for contextualized representations. The heuristic is designed to compare embedding methods and is *not* a bias measurement technique like WEAT or Log Probability Bias Score (Kurita et al., 2019).

### 3.1 WEAT

The Word Embedding Association Test (WEAT), proposed initially by Caliskan et al. (2017), defines the differential association of a target word $w$ with the attribute sets $A$ and $B$ as the difference in the means over $A$ and $B$ of the similarities of a target word with words in the attribute sets:

$$s(w, A, B) = [\mu_{a \in A}\text{sim}(w, a) - \mu_{b \in B}\text{sim}(w, b)],$$

where *sim* is any similarity metric (here the cosine similarity) between embeddings. The test statistic for a permutation test over target sets $X$ and $Y$, is then given as

$$S(X, Y, A, B) = \left[ \sum_{x \in X} s(x, A, B) - \sum_{y \in Y} s(y, A, B) \right]$$

We modify the original definition here to calculate the test statistic as below, allowing us to account for $X$ and $Y$ of different lengths.

$$S'(X, Y, A, B) = [\mu_{x \in X} s(x, A, B) - \mu_{y \in Y} s(y, A, B)]$$

For the null hypothesis stating that there is no statistically significant difference between the association of the two target groups with the two attribute groups, the $p$-value for the one-sided permutation test can thus be computed using this modified test statistic as

$$p = \text{Pr}[S'(X_i, Y_i, A, B) > S'(X, Y, A, B)]$$

where we compute the test statistic over all partitions $i$ of the combined set of target words and check if it is strictly larger than the test statistic from the original sets of target words.

We calculate Cohen's effect size $d$ as

$$d = \frac{S'(X, Y, A, B)}{\sigma_{w \in X \cup Y} s(w, A, B)}$$

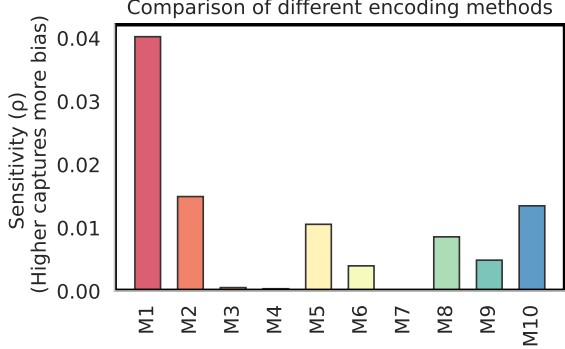

Figure 1: An example of the Korean language shows $M_5$ (average of embeddings from all hidden layers and considering average of subwords) as the contextualized embedding method with the highest sensitivity, a finding consistent across languages.

A negative effect size indicates an inverse association between the current target and attribute groups indicating that if we reverse the target sets while keeping the attribute groups the same, a positive association would be observed. A one-sided $p$-value greater than $0.95$ accompanying a negative effect size implies that the association is statistically significant after switching the targets.

### 3.2 Bias Sensitivity Evaluation

WEAT has traditionally been applied to uncover biases in static word embeddings like FastText and GloVe (Lauscher and Glavaš, 2019; Caliskan et al., 2022; Sesari et al., 2022). However, recent developments in the field have seen a significant shift towards contextualized embeddings, e.g. using transformer models such as BERT (Tan and Celis, 2019; Kurita et al., 2019; Silva et al., 2021). Various methodologies to extract contextualized word embedding representations from BERT exist, as partially explored by Devlin et al. (2019), but no single method consistently excels in representing words for detecting biased associations in WEAT across languages. Table 3 outlines the 10 encoding methods we examine.

In order to capture the extent to which an embedding method can discern alterations in word sets, influenced by translations into non-English languages and the inclusion of language-specific and gendered terms, we introduce a heuristic termed bias 'sensitivity'. This heuristic measures the average variance of pairwise cosine distances within word sets for each encoding approach. This rationale stems from the workings of the Word Embedding Association Test (WEAT), which computes

**Encoding methods**

bert layer 0 subword avg ($M_1$)/first ($M_2$)
bert 2nd last layer subword avg ($M_3$)/first($M_4$)
bert all layers subword avg ($M_5$)/first ($M_6$)
bert last layer CLS ($M_7$)
bert last layer subword avg ($M_8$)/first ($M_9$)
FastText ($M_{10}$)

Table 3: Cross-lingual analysis reveals $M_5$, $M_8$, $M_1$, and $M_{10}$ as the most sensitive for evaluating bias among the examined encoding methods.

pairwise cosine distances between groups of words, and posits that an encoding method exhibiting a higher variance in these distances would be more 'sensitive' to variations in these word sets.

Formally, Bias Sensitivity ($\rho_i$) for method $M_i$:

$$\rho_i = \frac{1}{6}\sum_{j=1}^{6}\overline{V}_{ij} = \frac{1}{6}\sum_{j=1}^{6}\left(\frac{1}{4}\sum_{k=1}^{4}V_{ijk}\right)$$

Here, $V_{ijk}$ is the variance of pairwise cosine distances for each set in each WEAT category, $j$ and $k$ denote the WEAT category and set index, respectively. The sensitivity is the mean of average variances across the WEAT categories.

Our findings indicate that the embedding layer from BERT, providing static word representations, exhibits the highest sensitivity, closely followed by FastText embeddings. However, as our analysis is centered on contextualized word embeddings, we select the representations from taking the mean of embeddings obtained from all hidden layers of BERT, considering the average of the subwords in each token (method $M_5$ from Figure 1), as this approach displays the subsequent highest sensitivity among the methods we chose for analysis. Results for the other three most sensitive methods are available in the Appendix A, B, C, and D. The CLS-based approach ($M_7$) is the only method where we consider the context of all the subwords and all the tokens in a phrase to get a single embedding from a contextual model like BERT without any subword averaging or hidden layer selections involved, but it is observed that this method usually has very low sensitivity scores indicating that it may not be suitable to use it for WEAT effect size calculations.

**Experimental Setting** We analyze cross-linguistic patterns for both existing WEAT categories and our newly introduced human-centered dimensions using two widely used multilingual models from Hugging Face (Wolf

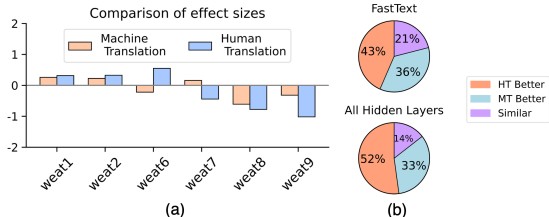

Figure 2: $M_5$ (average of embeddings from all hidden layers and considering average of subwords) and $M_{10}$ (FastText) exhibit higher effect sizes for Human Translations than Machine Translations across languages, as demonstrated for Korean.

et al., 2020) for extracting contextualized word embeddings - XLM-RoBERTa (Conneau et al., 2020) and DistilmBERT (Sanh et al., 2020). We employ our heuristic of bias sensitivity $\rho$ to guide our analysis. Our results[2] are compared with those from monolingual BERT models when available for a particular language for a broader perspective on language-specific biases. Additionally, we delve into Indian languages more extensively using an IndicBERT model (Kakwani et al., 2020).

### 3.3 The need for human translations

Our study compares machine translations (MT) provided by Google Translate with human translations (HT). To do so, we consider the same gender for grammatically gendered words and exclude the *language-specific* information in our dataset. This experiment aims to answer whether human translations are necessary to identify biases across different cultures or if MT systems are sufficient. Nevertheless, **our findings show that for any given WEAT category, our human translations more frequently yield larger effect sizes than MT, suggesting that sole reliance on MT may not suffice for bias evaluation across languages.**

Human-translated data produces a larger absolute effect size for all six categories in Korean. We also compare two of our most sensitive encoding methods in Figure 2 - static FastText and embeddings averaged across all hidden layers of BERT and find that regardless of encoding method or language, human translations provide a more comprehensive evaluation of bias, yielding larger absolute effect sizes than MT for the same category. MT systems are rarely perfect and may not understand the context in which a word is being used.

Another important example is from German, where gender bias has significant results when us-

---

[2]Data and Results are Available here : `https://github.com/iamshnoo/weathub`

ing Google Translate but when we use all of the *language-specific* as well as *gendered* words along with the human translations, the results become statistically insignificant. Therefore, we recommend utilizing human-annotated data for an accurate and fair assessment of bias across languages instead of solely relying on machine translation systems.

# 4 Results

We perform various experiments to understand how biases vary across languages, models, and embedding methods. In this section, we present the key takeaways from our experiments by highlighting statistically significant results, including some anecdotal examples from native speakers of a language aligned with our empirical findings in Appendix F. While we acknowledge that these examples might be influenced by the individual experiences of the selected annotator, it is crucial to understand that they are presented *only* as illustrative supplements to our statistically substantiated conclusions, not as a singular, potentially biased human validation of our results.

We also cover a case study of biases within the Indian context to explore common biased associations in India. The figures represent results for our chosen embedding method $M_5$ (mean of embeddings from all hidden layers considering the average of subwords in each token) as discussed in section 3.2. Results from the other embedding methods are available in the Appendix.

## 4.1 Cross-linguistic patterns in WEAT

Our exploration of bias across languages and encoding methods reveals intriguing variations for different WEAT categories (Figure 3).

**Contextualized embeddings often encode biases differently than static representations like FastText or the embedding layer in transformers.** For instance, when encoding methods involving BERT hidden layers are used, languages like Chinese and Japanese consistently display a pronounced positive bias. However, the same languages exhibit mixed or negative biases when using FastText or BERT's embedding layer. Urdu and Kurdish also follow a similar pattern, often revealing strong negative biases, though some categories and methods indicate low positive biases.

**Some languages consistently have positive biases while others have extreme effect sizes for specific categories.** For example, high-resource languages like Chinese and Telugu frequently demonstrate positive and significant biases across multiple categories. On the other hand, languages like Urdu, Tagalog, and Russian typically exhibit minimal positive or negative biases. Languages that lie in between, like Arabic, also make for interesting case studies because of some extreme effect sizes for particular WEAT categories, pointing to potential unique influences causing these outliers.

**Discussion** Inconsistencies across languages underscore the need for further exploration into the role of contextual understanding in the manifestation of biases. These cross-linguistic patterns in WEAT provide insights into the intricate dynamics of bias and emphasize the need for further nuanced investigation to comprehend the origin of these biases. We note, however, that these biases represent the findings of our experiments on querying language models in different languages and might be due to the biases (or lack thereof) expressed in those languages in the training corpora. Still, the training corpora for a given language must have originated from human-written text at some point, implying that its biases *might* directly reflect the language-level (or community-level) biases.

## 4.2 Multilingual vs Monolingual models

Models like DistilmBERT, which are trained on a diverse range of 104 languages, compared to monolingual models that focus on specific language corpora, can result in different findings about bias. Both types of models come with advantages and disadvantages. The selection between the two should be based on the individual requirements and context of the research or application.

**Culturally aligned biases can be discovered in monolingual models, sometimes in contrast to conclusions from a multilingual model, despite both results being statistically significant.** Take, for instance, the Thai context for WEAT 1, which relates flowers and insects to pleasantness and unpleasantness, respectively. Here, the multilingual model confirms the traditional association of flowers being pleasant and insects being unpleasant. Interestingly, a Thai-specific monolingual model robustly and statistically significantly contradicted this assumption (Appendix Table F.5). A web search shows multiple references claiming that Thais consider certain insects as culinary delicacies, indicating a preference for them. To verify this, we consulted with our Thai human annota-

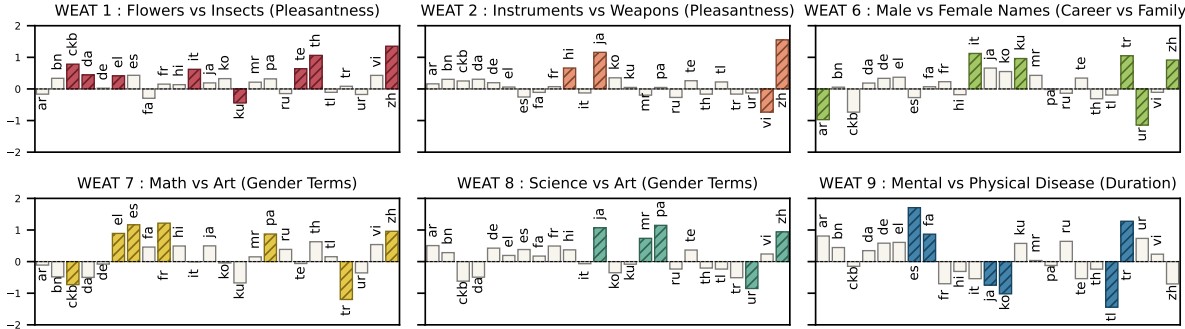

Figure 3: Effect size $d$ across languages for $M_5$ (average of embeddings from all hidden layers and considering average of subwords) in DistilmBERT. Significant results at 95% level of confidence are colored and shaded. Negative values of $d$ indicate reversed associations.

tor, who views eating insects as a common activity without explicitly associating it with liking or disliking. Therefore, in this particular instance, we favor the findings of the monolingual model.

**Biases are sometimes found in monolingual models even when these are not deemed statistically significant when querying a multilingual model.** For instance, the WEAT 7 category, which associates maths with male terms and arts with female terms, shows negative yet statistically insignificant effect sizes for Tagalog using multiple embedding methods in a multilingual model. However, when a monolingual model is used, these negative associations not only grow in effect size but also achieve statistical significance (Appendix Table F.6). Furthermore, for Turkish, the monolingual model identifies a significant positive bias for WEAT 1 and WEAT 2 (associating instruments with pleasantness and weapons with unpleasantness), which the multilingual model does not detect. On discussing with our Tagalog annotator, they agreed on some aspects of WEAT 7 showing female bias – they tend to associate math words more with females and art words with males, hence our findings perhaps indeed do reflect true social perceptions among Tagalog speakers.

**For high-resource languages, there is no significant difference in results from a monolingual model and a generic multilingual model.** For example, in Chinese, substantial and statistically significant biases for WEAT 1, 2, 7, and 8 are observed for both multilingual and monolingual models (Appendix Table F.8). We also find significant bias for WEAT 6 for the multilingual model, which is only detectable in the monolingual model for specific embedding methods.

**A model trained on a group of languages from the same language family does not always re-**

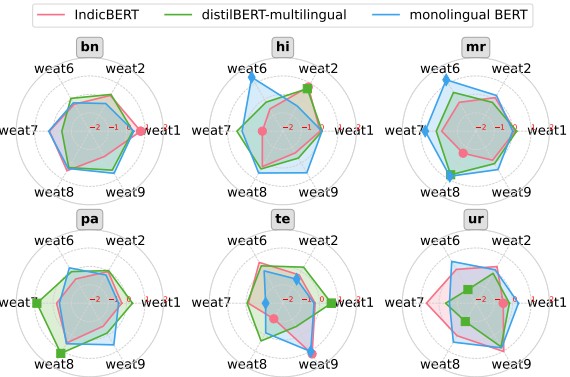

Figure 4: Monolingual models generally have larger effect sizes across languages and WEAT categories for $M_5$ (average of embeddings from all hidden layers and considering average of subwords). Markers (circles, diamonds, and squares) show significant results.

**flect biases identically to monolingual models for each such language.** This observation is evident from our case study involving six Indian languages. Figure 4[3] contrasts DistilmBERT with an IndicBERT model, pre-trained on twelve major Indian languages and monolingual models designed for each of the languages under consideration.

For Bengali, there is a high consistency across all models and WEAT categories. In languages like Hindi and Marathi, monolingual models exhibit the most significant effect sizes for WEAT 6 and 9, while DistilmBERT presents comparable performance with monolingual models for the remaining categories. Interestingly, the DistilmBERT model exhibits proficiency in lower-resource languages like Punjabi, uncovering larger effect sizes across most WEAT categories, except for WEAT 9. Similarly, DistilmBERT typically exhibits larger effect sizes for Telugu than the other models. However, the monolingual and IndicBERT models register

---

[3]Appendix Figure E.18 provides a higher resolution figure.

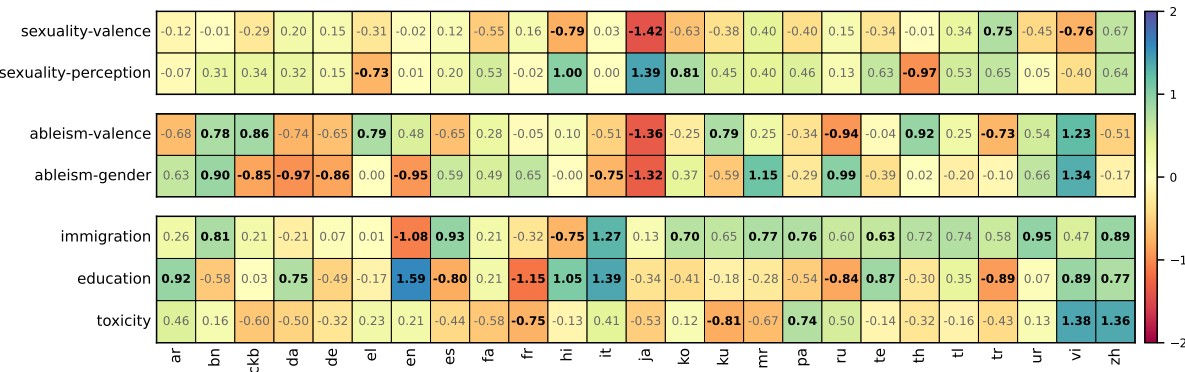

Figure 5: Effect sizes from $M_5$ (average of embeddings from all hidden layers and considering average of subwords) for contemporary biases in DistilmBERT; significant biases are indicated in bold, evidencing diverse language-specific trends across all dimensions.

larger effect sizes for Urdu than DistilmBERT.

Overall, monolingual models typically manifest effect sizes equal to or larger than those obtained from IndicBERT, exhibiting the lowest bias across various languages and WEAT categories. Furthermore, DistilmBERT can equal or surpass the effect size of the monolingual model in most languages, excluding Urdu. Multilingual models may not capture some language-specific biases that monolingual models can. Conversely, monolingual models yield results that multilingual models might overlook. These findings emphasize the complex role of model selection in analyzing linguistic biases.

### 4.3 Analysis of Human-Centered Biases

Our newly introduced dimensions of bias capture relevant modern social concepts, and we find statistically significant results from WEAT tests on DistilmBERT, XLM-RoBERTa, and also FastText, showing that these biases exist across languages.

**Ableism is present in language models, but it varies across languages.** We first assess ableism by examining the association of insulting terms with female terminology and the prevalence of disability-related terms with male terminology. Greek is an example where this phenomenon is evident in most encoding methods. Using various encoding methods, our experiments provide strong evidence of negative associations in German and Sorani Kurdish. In the case of Arabic, the embedding layer or CLS representation from the last layer of our multilingual DistilmBERT model shows fewer associations between female terms and insults. However, other encoding methods demonstrate a statistically significant positive bias in this category. Similar disparities across layers of the same model are also observed in Chinese.

When considering the first subword of each token, there is a significant negative bias in the ableism category. However, when using the average of all subwords, no significant bias is observed. We additionally measure word associations of ableism with pleasantness as measured by valence. Mostly similar trends are observed with differences in some of the languages. An interesting observation is that the effect sizes are sometimes reversed in directions across the two aspects of comparison for ableism. In Russian, for example, a strong positive bias is observed towards females but a strong negative association with unpleasant words.

**Considerable bias is observed in language models regarding education and immigration across various languages.** Hindi, Italian, Telugu, Chinese, and even English exhibit a significant positive bias, associating higher social status with education. However, there are exceptions to this trend. French, Punjabi, and Turkish display significant negative biases, suggesting that education is not universally linked to higher social status across cultures. Regarding immigration bias, our analysis explores whether terms related to immigration are more commonly associated with disrespectful words than respectful ones used for non-immigrants. A diverse range of languages from different cultures (Spanish, Italian, Kurmanji Kurdish, Turkish, Korean, Chinese, Bengali, Marathi, Punjabi, Urdu) demonstrate a significant bias for this category across various methods. These findings indicate that immigrants are more likely to be associated with disrespectful words in the embeddings from the multilingual DistilmBERT language model. However, English and Hindi exhibit negative biases, suggesting that immigrants are more likely to be respected in those languages.

| Language | Bias type | WEAT | Effect size (p-value) |
|---|---|---|---|
| Bengali | Religion | Adjectives vs Religion terms | -1.179 (0.996) |
| Hindi | Gender
Religion
Religion | Gendered entities vs Male, Female terms
Adjectives vs Religion terms
Adjectives vs Last Names | 0.824 (0.040)
0.965 (0.020)
1.414 (0.001) |
| Punjabi | Caste | Adjectives vs Caste Names | -1.216 (0.995) |
| Urdu | Gender
Caste
Occupation | Stereo Adjectives vs Male, Female terms
Adjectives vs Caste Names
Adjectives vs Urban/Rural occupations | -1.221 (0.994)
-1.107 (0.993)
0.997 (0.029) |

Table 4: Statistically significant results for India-specific bias dimensions of religion, caste, occupation and gender from $M_5$ (average of embeddings from all hidden layers and considering average of subwords) of DistilmBERT. Negative effect sizes are significant after reversing targets.

**Sexuality bias exists in multiple languages, whereas toxicity bias appears to be less prevalent.** In terms of the perception angle, Spanish, Persian, and Korean demonstrate substantial positive biases across multiple methods in WEAT 13, indicating a prevalent prejudice against LGBTQ+ communities. When using the multilingual DistilmBERT model, most languages do not exhibit significant biases for toxicity. However, upon comparing with the XLM-RoBERTa model, significant negative biases are predominantly observed, indicating that female terms are frequently associated with respectful words, while male terms tend to correlate with offensive ones. Additionally, we also measure associations with pleasantness and this valence angle also has similar trends across languages barring a few like Hindi and Japanese where the effect sizes are reversed in direction.

Figure 5 highlights these patterns of inherent biases that might exist in the structure of languages themselves or how they are used in different cultural contexts. However, it is crucial to note that these observations are based on the biases learned by the language model and may not necessarily reflect the attitudes of all speakers of these languages.

### 4.4 Biases in Indian languages

Malik et al. (2022) define some dimensions of bias common in India, for example, associating negative adjectives with the Muslim minority or discriminating based on caste. These biases are reflected strongly in Hindi when using word embeddings from a static model like FastText or contextual models like ElMo (Peters et al., 2018). However, for the other Indian languages we explored, namely English, Bengali, Urdu, Punjabi, Marathi, and Telugu, neither FastText nor embeddings from BERT have statistically significant results for most of these biases. However, we find some surprising patterns;

for example, embeddings for Urdu reverse gender bias by associating male terms with stereotypical female adjectives and vice versa.

The main takeaway from our study is that the stereotypes of bias in Hindi, the most prevalent language in India, are not equally reflected in other Indian languages. Recent work like Bhatt et al. (2022) provides further guidelines for exploring fairness across Indian languages. Table 4 lists the biases with significant effect sizes from our experiments. The complete set of results is available in our code repository.

## 5 Conclusion

We introduce a multilingual culturally-relevant dataset for evaluating 11 dimensions of intrinsic bias in 24 languages using target and attribute pairs for the WEAT metric. Beyond previously-studied dimensions of social bias like gender, we propose new dimensions of human-centered contemporary biases like ableism and sexuality to find strong evidence of bias in language models. We show that bias does not uniformly manifest across languages, but monolingual models do reflect human biases more closely than multilingual ones across multiple methods of extracting contextualized word embeddings. We also find that human translations are better suited for bias studies than automated (machine) translation ones. Finally, our case study on Indian languages reveals that biases in resource-heavy languages like Hindi are not necessarily found in other languages. WEAT, however, is an inconsistent metric for measuring biases, as indicated by a limited number of statistically significant results. In the future, we aim to develop metrics better suited for measuring biases in contextualized embeddings and generative models and explore their effects in real-world downstream applications.

## Limitations

- For most of the languages in our dataset WEATHub, we had access to at least two annotators for cross-verifying the accuracy of the human translations to determine if the translated words fit into the context of that particular WEAT category. However, for some languages, we only have one annotator per language. We plan to make our dataset available via open-source, opening it up to future crowdsourcing possibilities where we would look to get at least two annotators for each language where possible.

- While we have tried to cover as many languages from the global South as possible, we acknowledge that 24 languages are indeed a tiny proportion of the 7000 languages in the world, some of which do not even have text representations. Bias detection and mitigation in speech is one direction of research we plan to work on in the future. Another critical step is to facilitate crowdsourcing of WEATHub to cover low resource languages ("The Left-Behinds" and "The Scraping-Bys") from the taxonomy introduced in (Joshi et al., 2020) so that we can have fairer systems not just for high resource languages but for all languages.

- Among many other studies, Kurita et al. (2019) has previously shown how WEAT can be an unreliable metric for contextualized embeddings from transformer models. Silva et al. (2021) furthers this by showing how "debiasing" based on WEAT alone does not truly represent a bias-free model as other metrics still find bias. Badilla et al. (2020) provides a framework to simultaneously compare and rank embeddings based on different metrics. However, these studies reinforce that we need better metrics to study intrinsic biases in transformer models. We believe the target and attribute pairs we provide as part of WEATHub in multiple languages is an important step towards a better multilingual metric for evaluating intrinsic biases in language models.

## Acknowledgments

We are grateful to the anonymous reviewers for their constructive feedback. This work was generously supported by by the National Science Foundation under award IIS-2327143. Computational resources for experiments were provided by the Office of Research Computing at George Mason University (URL: https://orc.gmu.edu) and funded in part by grants from the National Science Foundation (Awards Number 1625039 and 2018631).

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

# A   Effect sizes for original WEAT categories from DistilmBERT and FastText

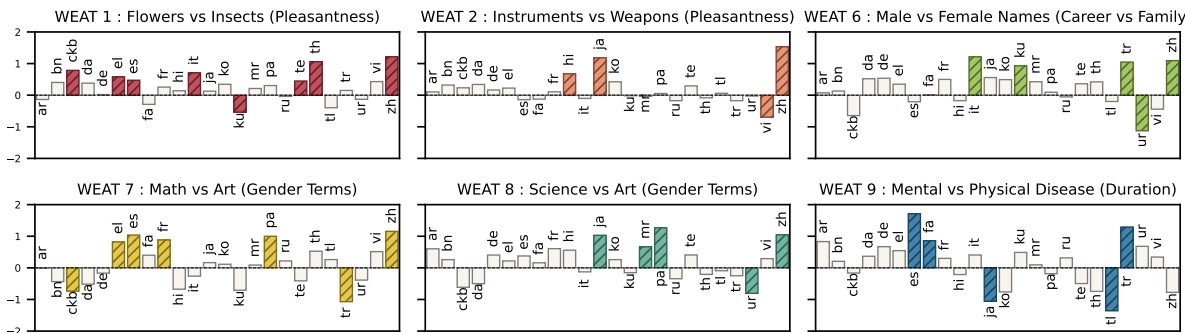

Figure 6: Effect size $d$ across languages for $M_1$ (embeddings from the static embedding layer and considering average of subwords) in DistilmBERT. Significant results at 95% level of confidence are colored and shaded. Negative values of $d$ indicate reversed associations.

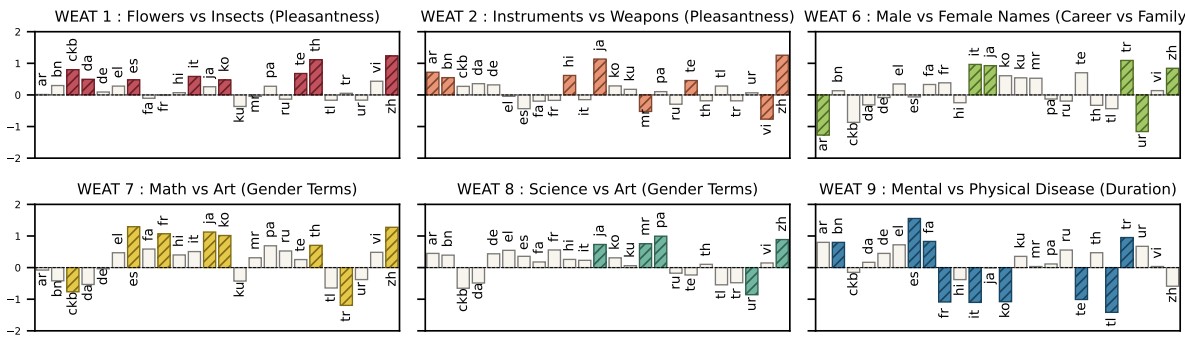

Figure 7: Effect size $d$ across languages for $M_8$ (embeddings of last hidden layer and considering average of subwords) in DistilmBERT. Significant results at 95% level of confidence are colored and shaded. Negative values of $d$ indicate reversed associations.

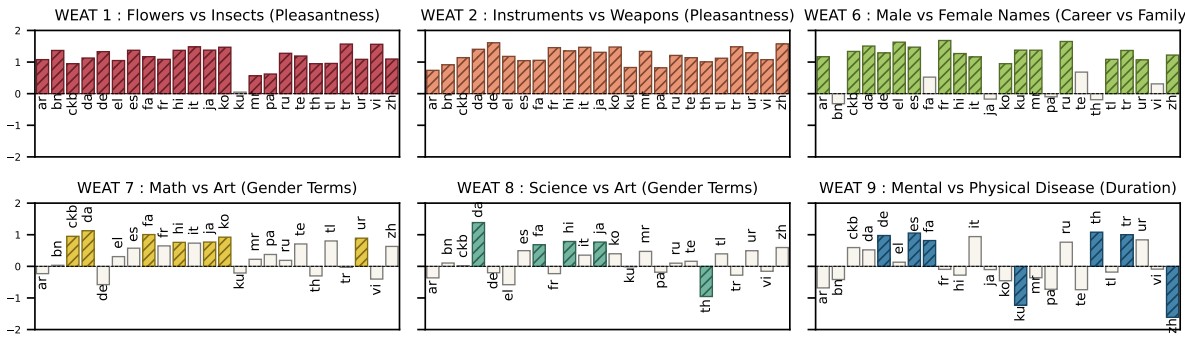

Figure 8: Effect size $d$ across languages for $M_{10}$ (FastText). Significant results at 95% level of confidence are colored and shaded. Negative values of $d$ indicate reversed associations.

## B Effect sizes for Human-Centered bias dimensions from DistilmBERT and FastText

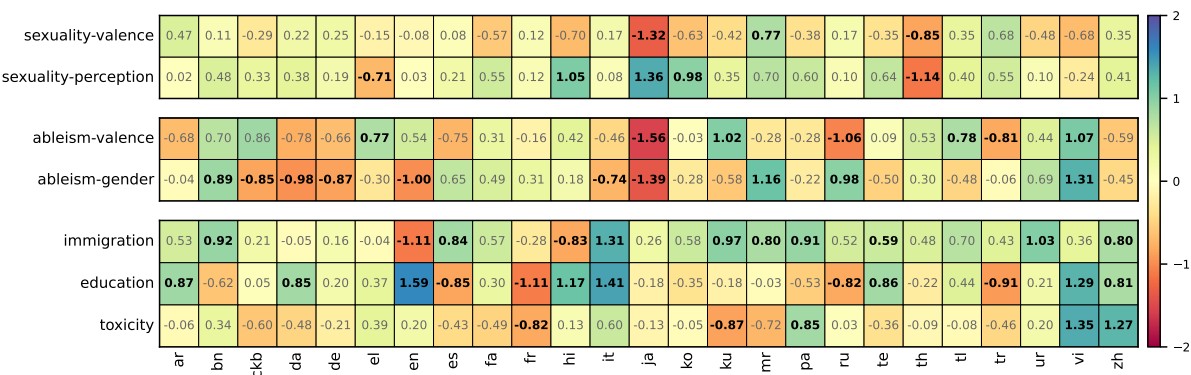

Figure 9: Effect sizes from $M_1$ (embeddings from the static embedding layer and considering average of subwords) for contemporary biases in DistilmBERT; significant biases are indicated in bold, evidencing diverse language-specific trends across all dimensions.

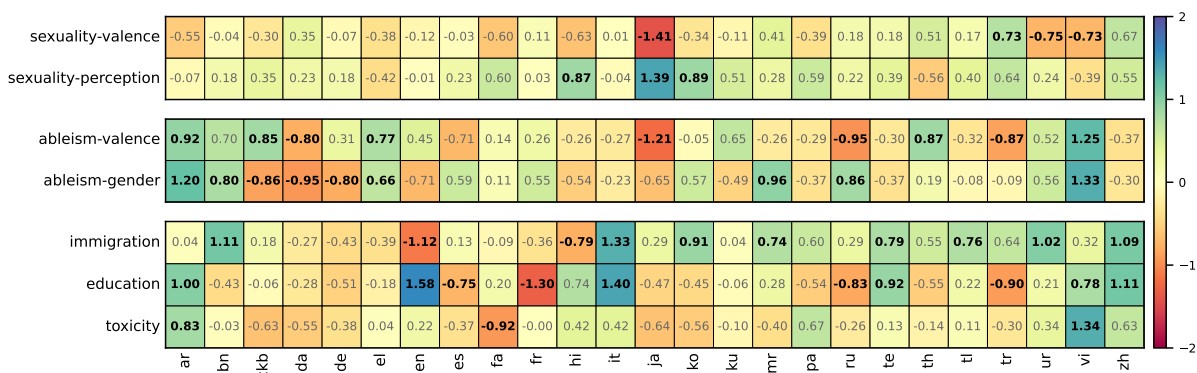

Figure 10: Effect sizes from $M_8$ (embeddings of last hidden layer and considering average of subwords) for contemporary biases in DistilmBERT; significant biases are indicated in bold, evidencing diverse language-specific trends across all dimensions.

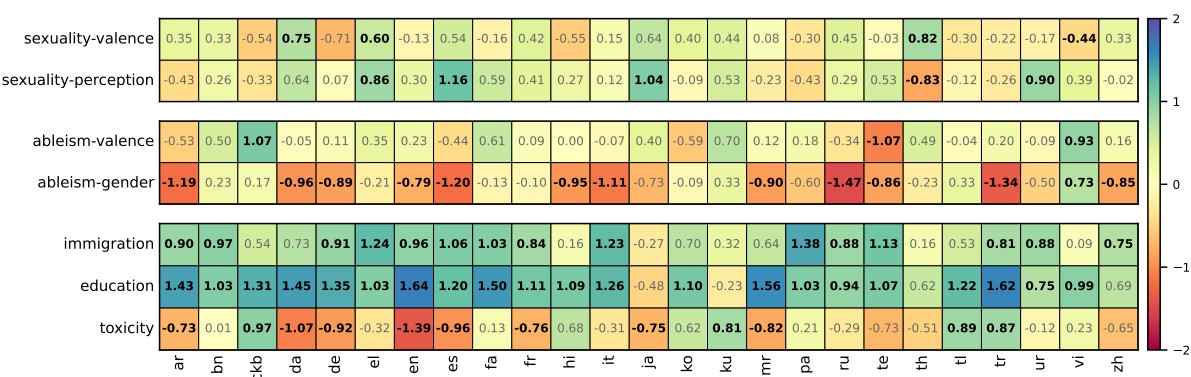

Figure 11: Effect sizes for contemporary biases in $M_{10}$ (FastText); significant biases are indicated in bold, evidencing diverse language-specific trends across all dimensions.

# C  Effect sizes for original WEAT categories from XLM-RoBERTa

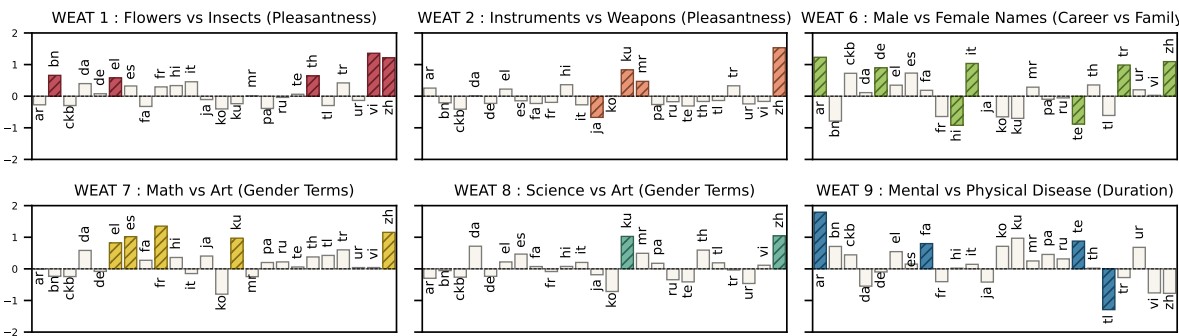

Figure 12: Effect size $d$ across languages for $M_1$ (embeddings from the static embedding layer and considering average of subwords) in XLM-RoBERTa. Significant results at 95% level of confidence are colored and shaded. Negative values of $d$ indicate reversed associations.

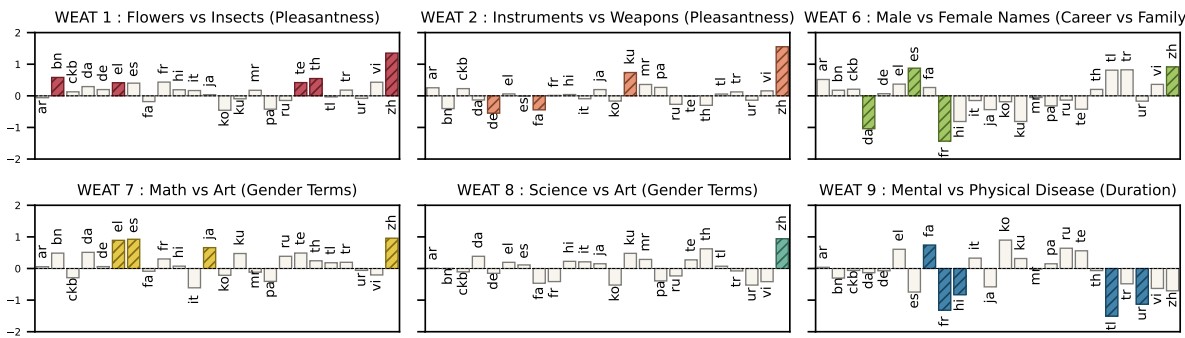

Figure 13: Effect size $d$ across languages for $M_5$ (average of embeddings from all hidden layers and considering average of subwords) in XLM-RoBERTa. Significant results at 95% level of confidence are colored and shaded. Negative values of $d$ indicate reversed associations.

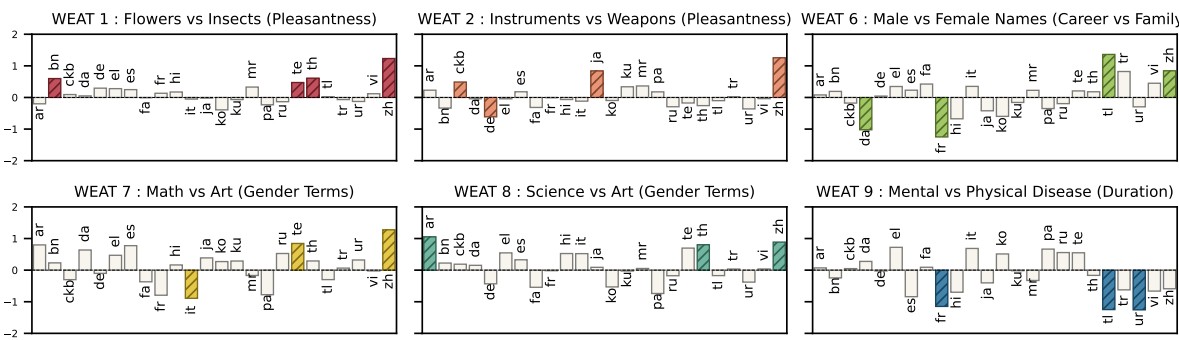

Figure 14: Effect size $d$ across languages for $M_8$ (embeddings of last hidden layer and considering average of subwords) in XLM-RoBERTa. Significant results at 95% level of confidence are colored and shaded. Negative values of $d$ indicate reversed associations.

# D Effect sizes for Human-Centered bias dimensions from XLM-RoBERTa

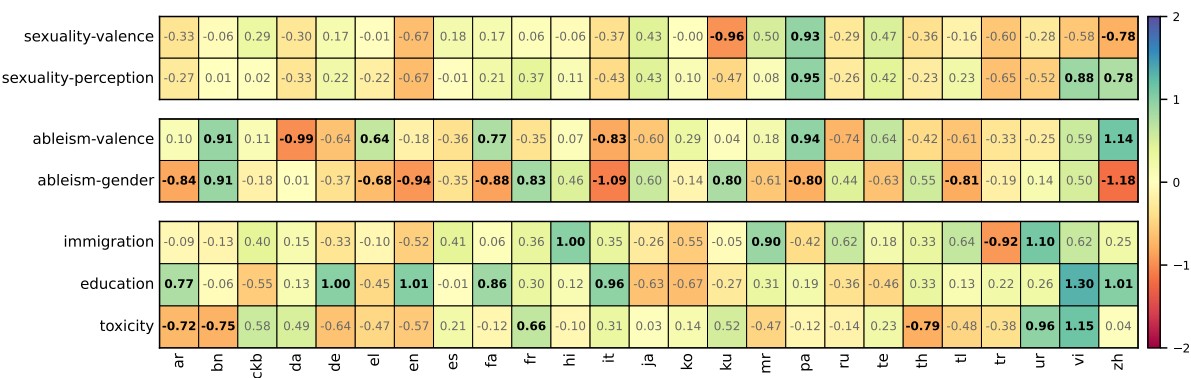

Figure 15: Effect sizes from $M_1$ (embeddings from the static embedding layer and considering average of subwords) for contemporary biases in XLM-RoBERTa; significant biases are indicated in bold, evidencing diverse language-specific trends across all dimensions.

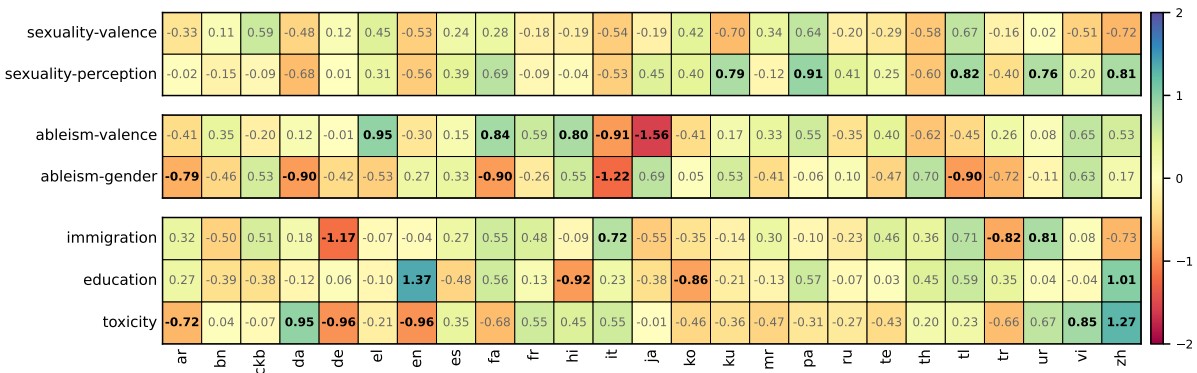

Figure 16: Effect sizes from $M_5$ (average of embeddings from all hidden layers and considering average of subwords) for contemporary biases in XLM-RoBERTa; significant biases are indicated in bold, evidencing diverse language-specific trends across all dimensions.

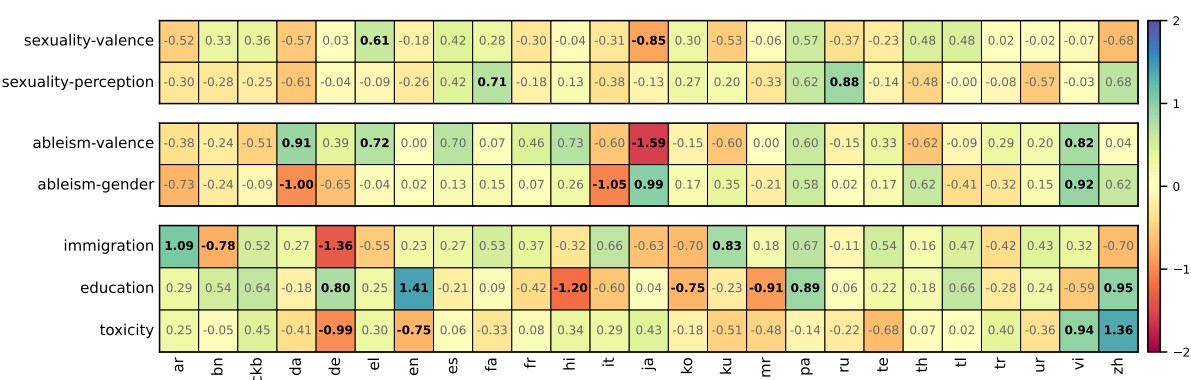

Figure 17: Effect sizes from $M_8$ (embeddings of last hidden layer and considering average of subwords) for contemporary biases in XLM-RoBERTa; significant biases are indicated in bold, evidencing diverse language-specific trends across all dimensions.

# E   Comparison of Monolingual and Multilingual models

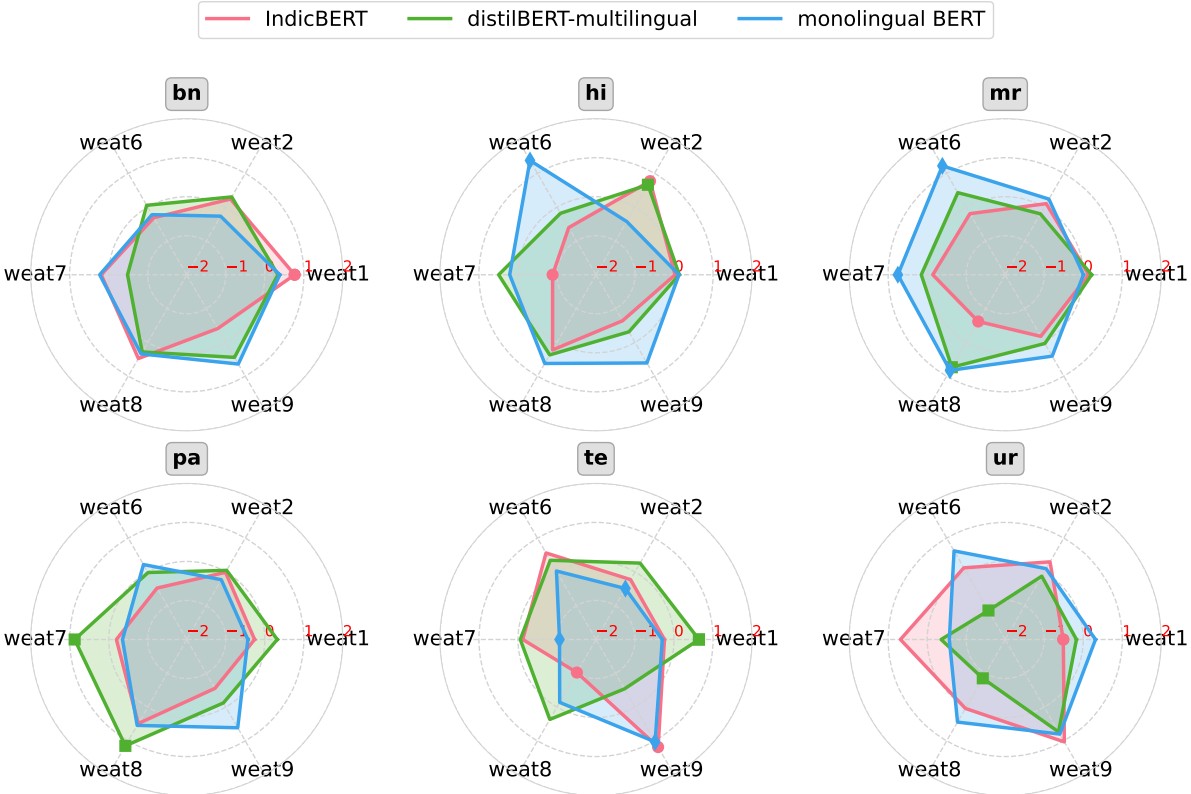

Figure 18: Monolingual models generally have larger effect sizes across languages and WEAT categories for $M_5$ (average of embeddings from all hidden layers and considering average of subwords). Markers show significant results.

## F Effect sizes and p-values for discussed results

| Method | Effect Size (p-value) |
|---|---|
| DistilmBERT | 1.062 (0.000) |
| Monolingual BERT | -0.629 (0.996) |

Table 5: Effect sizes and p-values for Thai WEAT 1 (Flowers:Insects::Pleasant:Unpleasant)

| Method | Effect Size (p-value) | Method | Effect Size (p-values) |
|---|---|---|---|
| $M_1$ | 0.262 (0.299) | $M_6$ | -0.240 (0.683) |
| $M_2$ | -0.293 (0.721) | $M_7$ | 0.838 (0.030) |
| $M_3$ | -0.028 (0.522) | $M_8$ | -0.645 (0.904) |
| $M_4$ | -0.198 (0.651) | $M_9$ | -0.119 (0.587) |
| $M_5$ | 0.155 (0.378) | $M_{10}$ | 0.803 (0.051) |

Table 6: Effect sizes and p-values for Tagalog WEAT 7 (Math:Art::Male Terms:Female Terms)

| Method | WEAT 1 | WEAT 2 |
|---|---|---|
| DistilmBERT | 0.084 (0.380) | -0.164 (0.733) |
| Monolingual BERT | 0.692 (0.003) | 0.533 (0.018) |

Table 7: Effect sizes and p-values for Turkish WEAT 1 (Flowers:Insects::Pleasant:Unpleasant) and WEAT 2 (Instruments:Weapons::Pleasant:Unpleasant)

| Method | WEAT 1 | WEAT 2 | WEAT 6 | WEAT 7 | WEAT 8 |
|---|---|---|---|---|---|
| DistilmBERT | 1.352 (0.000) | 1.552 (0.000) | 0.914 (0.033) | 0.959 (0.020) | 0.940 (0.023) |
| Monolingual BERT | 1.449 (0.000) | 1.511 (0.000) | 0.447 (0.194) | 1.111 (0.005) | 0.976 (0.016) |

Table 8: Effect sizes and p-values for different WEAT tests for Chinese