# OpenReview forum: "Global Voices, Local Biases: Socio-Cultural Prejudices across Languages"
_EMNLP/2023/Conference — EMNLP 2023 Main_

### Official Review · Reviewer_LVwA · 2023-08-04

**Soundness:** 4

**Excitement:**

4: Strong: This paper deepens the understanding of some phenomenon or lowers the barriers to an existing research direction.

**Missing References:**

N/a

**Paper Topic And Main Contributions:**

The authors expand the word embedding association test (WEAT) to 24 global south languages, with a particular focus on Indian languages,
and move from WEAT's focus on static embeddings to the contextualized represenations contained in language models through a behavioral
analysis of translation between the languages studied. Additionally, they compare machine translation into those languages against human translation.
Finally, they analyze culturally-specific embedded information in the model.

They use human annotators to correct machine translations of WEAT words into 24 languages, testing classes against attributes by comparison,
eg "flowers:insects::pleasant:unpleasant" or "math:art::male:female," identifying both innocuous and biased types of implicit word association.
Additionally, they test for ableism, toxicity, and sexuality, education, and immigration status biases.

They additionally test specifically for India-specific biases such as casteism and surname stereotypes that are unlikely to occur in non-Indian
languages, and compare their expression in Hindi (India's dominant local language) against other Indian languages such as Marathi and Telugu.

They introduce a bias sensitivity metric, the mean of the variances for the cosine similarities of embeddings between each set of classes (?) My
understanding of this was muddled until I checked back to the original WEAT paper, I would suggest the authors explain how the WEAT test works a
bit more in the intro to make the paper accessible to those outside the Ethical NLP space.

They find a few main results as well as many specific ones:

- contextualized word embeddings (eg higher layers in bert) do indeed express bias differently from static embeddings
- high-resourcedness makes representation biases more consistent in multilingual vs monolingual models as opposed to low resourced languages
- monolingual models tend to express more bias than multilingual ones (or would a better framing be "multilingual training mitigates some biases"?)
- Differential representation of gender, human-centric biases across Indian languages.

**I would really like to see the presentation issues I mentioned here addressed, but overall I think this paper should be published. By addressing the issues in reasons to reject, I think the authors can make their paper much more well-recieved and impactful.**

**Questions For The Authors:**

1. ~It seems to me like the Table 2 definition of Ableism is actually more a question of positive vs negative instead of gender? They use
"insult:disability term::male:female" whereas I'd suggest something like "insult:disability term::unpleasant:neutral/pleasant" as a way to
test for implict ableism in a model. Can you defend this choice or is this a typo?~ **Authors have added valence correlation in addition to gender correlation for ableism results. I hope that they will integrate this into the core findings and discussion, and provide their defense for the disability-gender correlation in the camera ready as they provided in their response.**
2. Are the sustained language-level positive/negative biases found in the contextualized embeddings (366-377) expressing language-level biases?
Or do you think they are genuinely reflective of differences in valence expressed in those languages in the training corpora directly (which I suppose is still a bias but would be addressed differently)?
3. Please consider reorganizing your presentation of methods and results, give the reader a clear understanding of what tangible things you will test (languages, models, etc) **on page 1** and then **make it clear which results are a part of that main experiment, and which are auxilliary findings**. See my "Confusing mixture" point in Reasons to Reject
4. Why not consider the India-specific biases in English?

**Reasons To Accept:**

Novel human-centered bias analysis. Great job widening the applicability of these bias classes by looking in the multilingual setting and including the bit focusing on India-specific biases.

Well-presented bolded findings in results section (apart from my gripes). They propose some hypotheses for why different observations are made between languages, although some discussion of them is thin.

Interesting results, novel generalization of WEAT to multilingual setting, novel bias classes.

Useful resource in WEAThub

**Reasons To Reject:**

**Post-rebuttal comments in bold: I believe the authors have satisfactorily responded to my reasons to reject and have updated my score.**

**the authors have responded to my complaints regarding organization and plan to reorder things in line with my and other reviewers suggestions. I think this will improve the quality of the paper considerably.** ~Intro, method sections are so vague in parts as to be confusing. From reading the intro alone I expected this would be a study of generative LM outputs and the differential expression of biases under translation into the languages, but in fact it's just a generalization of the WEAT test into 24 languages with culturally specific knowledge and a few tweaks to handle contextualized embeddings like BERT. That's fine, but the paper would be more readable if the abstract/intro were more upfront about this. In the methods section a lot of ink is spilled explaining p tests which I feel like is redundant for people who have taken stats 101 and useless for those who haven't. This space would be better spent doing things like:~

- ~listing all the languages and models they tested in the intro (see later point about "Confusing mixture")~
- ~giving background intuition on how WEAT works/is intended to work (eg. "WEAT purports to identify biases based on the relative proximity of word embeddings in latent space between words of negative valence and classes against which social bias is expressed in the real world" or something)~

**Authors plan to reorder results and introduce core elements earlier (eg in section 1 or 2) to make the results section less cluttered. This will improve readability and quality considerably.** ~Confusing mixture of discussion of the core result (bias in XLMBERT, which layers, which language etc) and auxilliary discussion points/findings along the way to the main result (need for human translation, some expressions are untranslatable, etc). I *love* that these extra points get discussed, it's just a frustrating read to have them mixed in seemingly at random (this is worst in secion 4, where they go from "Here are the HuggingFace models we tested" to "our study compares human translation and Google translate" to "these are the patterns we find". Please reorganize).~

**Authors have stated they will redo figures to be readable at a glance with descriptive labels rather than class numbers.** ~Figures are labeled with just WEAT classes, which leads to a lot of uninterpretable acronyms. ("weat 9" instead of "mental/physical disease bias")~

**Authors have addressed this (see questions for authors 1)** ~Framing of ableism as associating terms of disability with female is weird. See my first question for the authors.~

**Reproducibility:**

3: Could reproduce the results with some difficulty. The settings of parameters are underspecified or subjectively determined; the training/evaluation data are not widely available.

**Reviewer Confidence:**

3: Pretty sure, but there's a chance I missed something. Although I have a good feel for this area in general, I did not carefully check the paper's details, e.g., the math, experimental design, or novelty.

**Typos Grammar Style And Presentation Improvements:**

~Please provide information in the figure captions (or preferably, directly in the charts) about tests instead of just saying things like "M2" or "weat5". Figure 3 would be SO much better if instead of weat9 the caption said "Physical disease bias."~
**Authors have stated they will fix this in the camera-ready.**

---

> ### Author Rebuttal · Authors · 2023-08-28
>
> We thank you for taking the time to write a detailed review of our work! We will address the presentation issues that you have mentioned and are very happy to hear that you think this work should be published.
>
> **1. Reorganize introduction and method to introduce motive better:**
> - The first paragraph of the introduction mentions large language models to try to portray the problems of biases in these models as found by other studies. However, since our work does not focus on LLMs or generative outputs, we will modify this to make it more suited to our use case. The latter part of our intro focuses on explaining our contributions. Still, we understand how it might feel disconnected from the beginning discussion, and we will work to make it connect together better from a reader’s perspective.
>
> - Section 3.1  (Methods) briefly introduces the WEAT metric for readers unfamiliar with Ethics/Biases research. We included the discussion of $p$-values initially considering the same reader group. We like your suggestion that it will be helpful to have details of all the languages and the models in this space instead, along with an intuition for the metric. We will reorganize section 3.1 to accommodate this change. This will also solve the problem of having to start off the results section with a discussion of “Here are the models we tested,” as you mentioned.
>
> **2. Better flow in the results section:**
> - We appreciate your comment about how the results might appear to be mixed in randomly. Per your suggestions, we will reorganize section 4 and its subsections to have a clearer structure for the main results. The extra page for the camera-ready should help in managing that.
>
> **3. Figures are labeled with WEAT classes:**
> - We appreciate your feedback in helping us make our figures better. We will replace the usage of acronyms with the corresponding names wherever possible and update figure captions otherwise to make the figures more readable so that the dependencies on Tables 1,2 and 3 can be reduced.
>
> **4. Framing of ableism w.r.t. gender:**
> - After consulting with our annotators across all languages, we established our framing dimension. The consensus was that when discussing disabilities, terms such as "blind" or "visually impaired" tend to be used differently across genders in their respective languages.
> - Framing ableism in terms of valence might offer a perspective that's more intuitively graspable. We intend to undertake these experiments and incorporate the findings into the finalized version of our paper, in addition to the current framing.
>
> **Responses to your Questions:**
>
> **(1)** As discussed in point 4 of our reply above, we framed our dimensions based on the research we came across and the general opinion from conversations with native speakers in each of the languages. The resulting angle of comparing ableism w.r.t. gender is agreeably less intuitive than comparing ableism w.r.t. valence (pleasant: unpleasant). Running these valence experiments for the ableism dimension is straightforward because we already have the data for all languages. The main results of these experiments are given below. We will update the results in the appendix with the corresponding results in the final version of the paper.
>
> Ableism valence (Insult:Disability :: Unpleasant:Pleasant) results (effect sizes) for Method M5 (avg of all hidden layers of DistilmBERT and subword avg) bolded if significant at 95% level of confidence in alphabetical order of language codes similar to Figure 5 for all the 25 languages (including English) :
>
> | ar  | bn   | ckb  | da   | de   | el   | en   | es   | fa   | fr   | hi   | it   | ja    | ko   | ku   | mr   | pa   | ru   | te   | th   | tl   | tr   | ur   | vi    | zh   |
> |-----|------|------|------|------|------|------|------|------|------|------|------|-------|------|------|------|------|------|------|------|------|------|------|-------|------|
> | -0.68 | **0.78** | **0.86** | -0.74 | -0.65 | **0.79** | 0.48 | 0.65 | 0.28 | -0.05 | 0.10 | -0.51 | **-1.36** | -0.25 | **0.79** | 0.25 | -0.35 | **-0.94** | -0.04 | 0.92 | 0.25 | **-0.73** | 0.54 | **1.23** | -0.51 |
>
> **(2)** Lines (366-377) speak about the biases we find in languages by querying language models. We believe this might be due to the biases or the lack of them expressed in those languages in the training corpora. Still, the training corpora for a given language could have originated from human-written text at some point, implying that the biases in the training corpora might be directly reflective of the language level (or community-level) biases. We will update the wording of this paragraph to make this more obvious.
>
> **(3)** We will restructure the introduction, methods, and results to address the presentation issues that you have mentioned and make the paper more accessible to readers.
>
> **(4)** This is an interesting question because we could potentially consider local (country or culture) specific biases in the local language or any language that is not as commonly spoken within that locale. For example, we might choose to evaluate India-specific biases not just in English but also in the other languages in our dataset. Our work represents a first step towards comprehensive bias evaluation across languages, and the evaluation of country-specific biases in other languages is something that we are working on actively.  We will add the results for India-specific biases in English to the final version of our paper to make it more in line with all the other experiments.

---

### Official Review · Reviewer_LAHL · 2023-08-05

**Soundness:** 4

**Excitement:**

4: Strong: This paper deepens the understanding of some phenomenon or lowers the barriers to an existing research direction.

**Paper Topic And Main Contributions:**

The paper makes multiple contributions: extending the Word Embedding Association Test WEAT's word lists to multiple additional languages and dimensions, modifying its formulas to accomodate these additions and employing this new resource to perform bias analysis in several settings.

**Questions For The Authors:**

a) Why is the BERT's CLS token considered as an encoding for the words? Unless I'm missing something, this token will encode the entire sentence/input rather than only the relevant word.

b) How/Why were the words/terms for the dimensions toxicity, ableism and sexuality chosen?

**Reasons To Accept:**

a) Multiple contributions and extensive experiments

b) Interesting, relevant results

c) Mostly well-written paper

**Reasons To Reject:**

a) Additional Attribute-Target pairs for WEATHub seem partially unintuitive:
The bias dimensions toxicity and ableism are characterised by the association between female/male terms and offensive/respectful or insult/disability words, respectively. While I can somehow understand this in the case of toxicity (females may more often be the target of toxicity, even thoough I'm not entirely conviced this is what's supposed to be shown here), I don't really understand it for the ableism dimension. Similarly, the sexuality dimension tests the association between LGBTQ+/straight words and prejudice/pride, which doesn't seem intuitive to me.

b) Validation of results:
The results of the bias detection are not sufficiently validated. While the findings are interesting, I'm not convinced that the biases discovered by the method are actually correct. In some cases, they are validated anecdotally by asking one of the native speakers serving as annotators, but this is not convincing enough to me - I would not be comfortable deciding whether there is generally a bias with regards to a group of people, objects, etc. in my native language.

Overall, the paper makes interesting contributions and will be impactful due to its resources. However, I think that the issues raised above should be fixed before acceptance.

**Reproducibility:**

4: Could mostly reproduce the results, but there may be some variation because of sample variance or minor variations in their interpretation of the protocol or method.

**Reviewer Confidence:**

3: Pretty sure, but there's a chance I missed something. Although I have a good feel for this area in general, I did not carefully check the paper's details, e.g., the math, experimental design, or novelty.

---

> ### Author Rebuttal · Authors · 2023-08-28
>
> Thank you for your valuable feedback on our research! We appreciate your recognition of the impact of our paper and are glad to see that you find the contributions interesting and well-written.
>
> **(a) Additional Target Attribute Pairs for WEATHub:**
> - We apologize for any confusion caused by our presentation of the human-centered contemporary biases. Our methodology for selecting the proposed dimensions, was fourfold:
>
>     **(1)** We start with insights from intersectional biases research (lines 161-163 have relevant citations).
>
>     **(2)** Then we engage in discussions with native speakers of the 24 languages establishing a foundational consensus on bias dimensions across linguistic communities.
>
>     **(3)** Subsequently, we formulated the corresponding target/attribute pairs after weighing different options for naming them.
>
>     **(4)** Finally, we opted for a naming convention that we felt best depicted these biases.
>
> - Representing biases across cultures under a single umbrella is a nuanced task. Our work represents a first step in this direction, and we will explore other combinations of target/attribute pairs.
> - Your understanding of the toxicity dimension is exactly as we intended. For sexuality bias, our objective was to highlight biases against LGBTQ+ groups and for ableism bias we designed it to spotlight distinct challenges faced by women with disabilities.
> - We understand that these framings need clearer elaboration, so we’ll incorporate clarifications and new experiments on valence in the final version of our paper on acceptance. With word lists for valence already available in our data for all the 24 languages, these experiments would enhance our paper and align it with recent research on valence like https://arxiv.org/pdf/2307.03360.pdf.
> - We already have the results for the main paper experiment setting for ableism and sexuality corresponding to Figure 5 as below. The appendix results for the same would be updated in the final version of the paper.
>
> **(1)** Ableism valence (Insult:Disability :: Unpleasant:Pleasant) results (effect sizes) for Method M5 (avg of all hidden layers of DistilmBERT and subword avg) bolded if significant at 95% level of confidence in alphabetical order of language codes similar to Figure 4 for all the 25 languages (including English) :
>
> | ar  | bn   | ckb  | da   | de   | el   | en   | es   | fa   | fr   | hi   | it   | ja    | ko   | ku   | mr   | pa   | ru   | te   | th   | tl   | tr   | ur   | vi    | zh   |
> |-----|------|------|------|------|------|------|------|------|------|------|------|-------|------|------|------|------|------|------|------|------|------|------|-------|------|
> | -0.68 | **0.78** | **0.86** | -0.74 | -0.65 | **0.79** | 0.48 | 0.65 | 0.28 | -0.05 | 0.10 | -0.51 | **-1.36** | -0.25 | **0.79** | 0.25 | -0.35 | **-0.94** | -0.04 | 0.92 | 0.25 | **-0.73** | 0.54 | **1.23** | -0.51 |
>
>
>    **(2)** Sexuality valence (LGBTQ+:Straight :: Unpleasant:Pleasant) results (effect sizes) for Method M5 (avg of all hidden layers of DistilmBERT and subword avg) bolded if significant at 95% level of confidence in alphabetical order of language codes similar to Figure 4 for all the 25 languages (including English):
>
> | ar   | bn   | ckb  | da   | de   | el   | en   | es   | fa   | fr   | hi   | it   | ja    | ko   | ku   | mr   | pa   | ru   | te   | th   | tl   | tr   | ur   | vi   | zh   |
> |------|------|------|------|------|------|------|------|------|------|------|------|-------|------|------|------|------|------|------|------|------|------|------|------|------|
> | -0.12 | -0.01 | -0.29 | 0.20 | 0.15 | -0.31 | -0.02 | 0.12 | -0.55 | 0.16 | **-0.79** | 0.03 | **-1.42** | -0.63 | -0.38 | 0.40 | -0.40 | 0.15 | -0.34 | -0.02 | 0.34 | **0.75** | -0.45 | **-0.76** | 0.67 |
>
> **(b) Validation of results:**
> - We regret the ambiguity arising from the presentation of our results. Any observations highlighted in bold in our results are _statistically verified_ outcomes from WEAT, confirmed as significant through $p$-value testing. These results can be cross-referenced in our supplementary code or via the anonymous link. Figures 3, 4 and 5 also refer to these significant biases and their captions explicitly mention this.
> - The anecdotal examples (lines 410-424 and 442-446) are intended to highlight how the language-specific biases we statistically identified in language models manifest in both our annotator (a native speaker of the said language) and web searches. While we acknowledge that these examples might be influenced by the individual experiences of the selected annotator, it's crucial to understand that they are presented merely as illustrative supplements to our statistically substantiated conclusions, not as a singular, potentially biased human validation.
> - We will clarify these points in our paper in section 4, so as to avoid potential confusion.
>
> **Responses to your Questions:**
>
> **(a) BERT CLS:**
> - As you rightly point out, it is common to use “contextualized embedding” to refer to embeddings from BERT in a sentential context. Instead, we are directly providing the word (or phrase in some cases) as input for the embedding method.
> - How does this make the resulting embedding different from the 7 other BERT hidden layer-based approaches we have compared in our work? In all the other methods, we choose either the first subword or the average of the subwords. If we encounter phrases in the input, we average the embeddings across the different tokens in the phrase after choosing a corresponding subword strategy. The CLS-based approach is the only method where we consider the context of all the subwords and all the tokens in a phrase to get a single embedding from a contextual model like BERT without any averaging or selections involved.
> - We will clarify this detail in the paper in section 3.2 to avoid any potential confusion.
>
> **(b) Words for Toxicity, Ableism, Sexuality:**
> - Lines 189-198 provide a brief overview of our approach to creating the list of words for each of the new bias dimensions.
> - To expand on that, as discussed in our response (about additional Target/Attribute Pairs in WEATHub) above, Firstly, we crafted potential names for the dimensions and the corresponding target/attribute class names. Subsequently, we devised an initial set of English terms for each category by amalgamating information from online platforms such as Wikipedia and the studies referenced in lines 161-163. This preliminary set was refined based on discussions with our annotators and by querying various open-source and closed-source large language models. The ultimate selection of words signifies a shared understanding and agreement between our team and annotators.
> - As discussed before, we will include experiments in valence measurements for these dimensions to keep our work on par with contemporary research and make it more intuitive.
>
> We hope that our explanations above clarify any questions regarding the soundness of our work (as reflected by the low soundness score). If you feel that there are more fundamental issues, please do get back to us with more questions!

---

### Official Review · Reviewer_L4qs · 2023-08-06

**Soundness:** 3

**Excitement:**

3: Ambivalent: It has merits (e.g., it reports state-of-the-art results, the idea is nice), but there are key weaknesses (e.g., it describes incremental work), and it can significantly benefit from another round of revision. However, I won't object to accepting it if my co-reviewers champion it.

**Missing References:**

* Not penalizing the paper for this, since it is contemporaneous, but [1] and several papers cited in [1] are relevant for future iterations of this work.

[1] - https://arxiv.org/abs/2305.11242

**Paper Topic And Main Contributions:**

The objective of this paper is to extend WEAT, the classic test for bias in word embeddings, to non-Western/European languages and locales. The paper has several contributions to advance this goal : first, a dataset of target and attribute pairs (like the original WEAT) are created for a subset of previously identified bias dimensions + 5 new dimensions. This dataset is created by semi-automatically translating existing English data to these languages. Second, the WEAT is adapted for multilingual settings by capturing multi-word translations of lexical items in English as well as by identifying a new metric to measure bias. An extensive evaluation of various pre-trained masked language models is carried out on this dataset and several observations about the models and languages studied are presented.

**Reasons To Accept:**

* Extensive evaluation of WEAT to 24 languages, including several languages not commonly studied in the fairness literature. The extension to human-centered contemporary biases is particularly noteworthy. Most of the evaluation is very sounds and backed by a large set of empirical results using the mechanisms and data developed in this work.

**Reasons To Reject:**

* The 24 languages mentioned in the paper are not explicitly spelled out anywhere, except for figures in the appendix (and that too using language codes). This is quite a glaring omission.
* It is difficult to judge the quality of the contribution of the dataset, since the paper does not include a single example, and the anonymous link does not work.
* The bias sensitivity metric is a little difficult to interpret, since it is not contextualized with respect to previous metrics (e.g. those by Kurita et al.) in the results. While I understand what the metric aims to do, the justification of creating a new metric is unclear.
* Some observations in Section 4.3 are not backed up very rigorously. For instance, the conclusion in line 406 is driven by a web-search + conversation with a single annotator. Similarly, the conclusion in line 426 is also based on a single annotator. It is unclear to what extent this is influenced by selection bias.

**Reproducibility:**

2: Would be hard pressed to reproduce the results. The contribution depends on data that are simply not available outside the author's institution or consortium; not enough details are provided.

**Reviewer Confidence:**

4: Quite sure. I tried to check the important points carefully. It's unlikely, though conceivable, that I missed something that should affect my ratings.

---

> ### Author Rebuttal · Authors · 2023-08-28
>
> Thank you for the insightful feedback on our work! We appreciate your acknowledgment of our comprehensive evaluations of WEAT in 24 less-represented languages and our efforts to extend this to address contemporary, human-centered biases. We assure that the main concerns raised are fixable and we will fix them in the final version of the paper.
>
> **Language names not explicitly mentioned:**
> - We apologize for this error and will explicitly mention the language codes in all figures and appendix material in the final version of the paper upon acceptance.
> - The languages (in alphabetical order of language codes) are: Arabic (ar), Bengali (bn), Sorani Kurdish (ckb), Danish (da), German (de), Greek (el), Spanish (es), Persian (fa), French (fr), Hindi (hi), Italian (it), Japanese (ja), Korean (ko), Kurmanji Kurdish (ku), Marathi (mr), Punjabi (pa), Russian (ru), Telugu (te), Thai (th), Tagalog (tl), Turkish (tr), Urdu (ur), Vietnamese (vi), Chinese (zh)
>
> **Anonymous link not working:**
> - We apologize for any inconvenience caused due to the anonymous GitHub link not working, which might be due to outages on the external server it is hosted on.
> - The complete dataset, associated code for the experiments, and the results are provided in a zip file accompanying the original submission (which can be downloaded from the Supplementary Materials section of our submission). We kindly ask you to refer to the provided folder to access examples from our dataset, in case the anonymous link fails to work again.
>
> **Bias sensitivity metric:**
> - We acknowledge the concerns regarding our metric’s description. However, the bias sensitivity metric is not used to measure biased associations in word embeddings. Instead, it is a heuristic for comparing encoding methods. On the other hand, metrics like WEAT and the work by Kurita et al. (LPBS or Log Probability Bias Score) are metrics directly used for measuring biases in word embeddings. Thus, we do not need to and cannot compare them with our bias sensitivity metric.
> - We will clarify these details in the paper by reorganizing section 3.2 and section 3.1 to explain the significance of our metric better.
>
> **Observations in section 4.3:**
> - We regret the ambiguity arising from the presentation of our results. Any observations highlighted in bold in our results are _statistically verified_ outcomes from WEAT, confirmed as significant through $p$-value testing. These results can be cross-referenced in our supplementary code or via the anonymous link. Figures 3, 4 and 5 also refer to these significant biases and their captions explicitly mention this.
> - Following lines 406 and 426, we have provided anecdotal examples. These are intended to highlight how the language-specific biases we statistically identified in language models manifest in both our annotator (a native speaker of the said language) and web searches. While we acknowledge that these examples might be influenced by the individual experiences of the selected annotator, it's crucial to understand that they are presented merely as illustrative supplements to our statistically substantiated conclusions, not as a singular, potentially biased human validation.  We will clarify these points at the beginning of the section, so as to avoid potential confusion.
>
> **Contemporaneous citations suggested:**
> - Thank you for the helpful suggestions. We will include them in the paper to improve our quality of references even more.
>
> We hope that our explanations above clarify any questions regarding the soundness of our work (as reflected by the low soundness score). If you feel that there are more fundamental issues, please do get back to us with more questions.

---

### Meta-Review · Area_Chair_Y6Z6 · 2023-09-19

**Recommendation:** 5

**Metareview:**

This paper expands Word Embbedding Association Test (WEAT) by translating from English to 24 languages. Overall, this paper makes good contributions by 1) proposing a new resource for bias and fairness evaluation through multiple dimensions, 2) empirical results on pretrained masked language models on wide range of languages. During the rebuttal, multiple concerns were raised about presentation clarity (e.g., flows of the results section), but we believe this is an easy fix to make the paper stronger. In summary, the AC is looking forward to the proposed datasets to be contributed to the community.

---

### Decision · Program_Chairs · 2023-10-07

**Decision:**

Accept-Main

**Comment:**

This paper expands Word Embbedding Association Test (WEAT) by translating from English to 24 languages. Overall, this paper makes good contributions by 1) proposing a new resource for bias and fairness evaluation through multiple dimensions, 2) empirical results on pretrained masked language models on wide range of languages. During the rebuttal, multiple concerns were raised about presentation clarity (e.g., flows of the results section), but we believe this is an easy fix to make the paper stronger. In summary, the AC is looking forward to the proposed datasets to be contributed to the community.